# Towards Robust Multimodal Learning via Adaptive Model Assembly

## Abstract

Adversarial fine-tuning is a widely used strategy to enhance the robustness of vision-language pre-trained models (VLPs) such as CLIP, ALBEF, and TCL. Traditional methods, however, typically fine-tune a single static model to defend against a specific attack type, limiting their ability to generalize to diverse or unseen adversarial threats. To address this, we propose Multimodal Adaptive Adversarial Fine-tuning (MAAF), a novel framework that achieves robust multimodal learning by adaptively assembling input-conditioned model parameters at inference time. MAAF starts from a shared base model and learns multiple defense vectors, which are dynamically fused through a lightweight, input-aware generation network to produce robust, sample-specific model parameters. This adaptive assembly allows the model to resist a wide range of adversarial attacks without retraining. Extensive experiments on standard vision–language benchmarks show that MAAF substantially enhances adversarial robustness while preserving clean accuracy, consistently outperforming existing fine-tuning methods. The results also provide insights into the distribution of defense vectors, the importance of adaptive fusion, and the optimal number of vectors for achieving a balance between robustness and stability. Code is available at https://anonymous.4open.science/r/MAAF-63FC.

## 1 Introduction

The proliferation of powerful vision-language pre-trained models (VLPs) such as CLIP (Radford et al., 2021), ALBEF (Li et al., 2021), and TCL (Yang et al., 2022) has revolutionized tasks requiring a deep, integrated understanding of visual and textual data. These models learn cross-modal representations from large-scale datasets and serve as foundations for a range of downstream tasks, including image-text retrieval (Liu et al., 2021), visual entailment (Song et al., 2022), and multimodal reasoning (Zhang et al., 2025). However, their growing deployment has exposed a critical vulnerability: a susceptibility to adversarial attacks. By introducing subtle but malicious perturbations to images or text, adversarial examples can significantly degrade model performance, even though the perturbations are often imperceptible to humans.

To counter adversarial threats in multimodal models, prior research has primarily relied on adversarial training strategies aimed at enhancing model robustness. However, these approaches typically train a single static model to withstand a specific type of attack—whether visual, textual, or cross-modal perturbations. For example, Co-Attack (Zhang et al., 2022) targets vulnerabilities in joint representations, while SA-Attack (He et al., 2023) and SGA (Lu et al., 2023a) employ data augmentation techniques to simulate adversarial scenarios. Despite their effectiveness in controlled settings, such single-model fine-tuning approaches often result in brittle defenses, failing to generalize beyond the particular attack types they were trained on. This limitation becomes especially pronounced in real-world scenarios, where adversarial strategies are diverse, adaptive, and often unseen during training.

To address these challenges, we propose Multimodal Adaptive Adversarial Fine-tuning (MAAF), a defense framework that generates input-specific protection at inference time. Unlike static approaches, MAAF dynamically adapts its parameters for each incoming sample. As illustrated in Figure 1, it learns a set of specialized defense vectors and employs a lightweight controller network to fuse them with a base model, creating a customized model instance for each input and activat-

Figure 1: Procedure of MAAF: Adaptively assembles a model per input by combining a base model $\theta_0$ with $K$ defense vectors $\{\Delta\theta_1, \cdots, \Delta\theta_K\}$ for robust, attack-specific defense.

ing the most relevant defense mechanisms. This input-adaptive design delivers robust and flexible protection against a wide range of multimodal adversarial attacks, including those not encountered during training.

Our main contributions are summarized as follows:

- **MAAF framework:** We propose Multimodal Adaptive Adversarial Fine-tuning (MAAF), which learns a set of defense vectors and dynamically fuses them to generate input-specific defense models, achieving strong robustness against diverse adversarial attacks. Additionally, we introduce a random multimodal attack that unifies existing unimodal and multimodal attack strategies, enabling comprehensive and consistent evaluation.

- **Bi-level optimization:** We develop a bi-level training scheme that jointly optimizes the defense vectors and the input-conditioned fusion network, facilitating instance-aware model adaptation.

- **Extensive evaluation:** Experiments on standard VLP benchmarks demonstrate that MAAF significantly improves adversarial robustness while maintaining high clean accuracy, and reveal insights into defense vector distributions, the critical role of adaptive fusion, and the optimal number of vectors for balanced robustness and stability.

## 2 RELATED WORK

**Vision-Language Pretraining Models (VLPs).** Early Vision-language pretraining models (VLPs), such as CLIP (Radford et al., 2021), ALIGN (Jia et al., 2021), and ALBEF (Li et al., 2021), adopt a dual-stream architecture, where images and texts are encoded separately using modality-specific encoders (e.g., Vision Transformers and BERT-like models). These models are typically trained with contrastive objectives or matching losses, which encourage paired image-text representations to align in a shared embedding space while separating unrelated pairs. Recent vision–language models (VLPs) have increasingly adopted unified, autoregressive architectures that fuse visual and textual inputs into a single generation pathway. Models such as BLIP (Li et al., 2022), Flamingo (Alayrac et al., 2022), GPT-4V (Yang et al., 2023), Gemini (Team et al., 2023), and Qwen2-VL (Wang et al., 2024) exemplify this trend. While the exact architectures of GPT-4V and Gemini remain undisclosed, they are widely believed to follow similar unified designs. In contrast, models like BLIP and Flamingo explicitly incorporate lightweight fusion modules—such as Q-Former and Perceiver Resampler—to connect visual encoders with large language models. Qwen2-VL adopts a comparable strategy by integrating visual inputs into a decoder-only language model through dynamic visual tokenization and modality-aware positional encoding.

**White-box Adversarial Attacks.** Despite the impressive performance of VLPs across a variety of multimodal downstream tasks, their vulnerability to adversarial attacks has emerged as a significant challenge. White-box adversarial attacks refer to scenarios where the attackers has full access to the internal details of the target model during the generation of adversarial samples. This includes access

to the model's architecture, parameter weights and the gradients of the loss function. Early studies primarily concentrated on unimodal attacks, such as PGD (Projected Gradient Descent) Madry et al. (2017) and BERT-Attack Li et al. (2020), demonstrating that even minor perturbations to visual inputs could substantially impair model performance. Subsequently, more advanced strategies emerged, exploiting the multimodal nature of VLPs. For example, Co-Attack Zhang et al. (2022) seeks to maximize the feature-space divergence between original and perturbed data across modalities, while approaches such as SGA Lu et al. (2023a) and SA-Attack He et al. (2023) utilize data augmentation techniques to disrupt the intrinsic cross-modal alignments within the model.

**Adversarial Robustness.** Vision–language pretraining (VLP) models are susceptible to adversarial perturbations on one or both modalities, which can lead to incorrect outputs Szegedy et al. (2014) and limit their reliability in real-world applications. Early defenses have largely focused on unimodal robustness. For instance, TeCoA Mao et al. (2022) uses PGD-based adversarial training with contrastive learning to align robust visual representations, while FARE Schlarmann et al. (2024) applies unsupervised adversarial fine-tuning by minimizing the distance between embeddings of clean and adversarial samples. Nevertheless, these unimodal defenses often fail against multimodal attacks, which exploit cross-modal dependencies to disrupt joint representations. Such methods typically improve robustness only against a specific attack with fixed settings and lack the flexibility to generalize across diverse attack types or configurations.

## 3 PRELIMINARIES

Let $\mathcal{M}_\theta$ denote a vision-language pretrained model parameterized by $\theta$, which takes an image $x_{\mathrm{v}}$ and a text $x_{\mathrm{t}}$ as input to produce a task-specific output: $y = \mathcal{M}_\theta(x_{\mathrm{v}}, x_{\mathrm{t}})$, where $y$ may represent classification logits (e.g., VQA), token sequences (e.g., captioning), similarity scores (e.g., retrieval), or structured outputs (e.g., grounding). We denote the modality-specific embeddings as $f_\theta^{\mathrm{v}}(x_{\mathrm{v}})$ and $f_\theta^{\mathrm{t}}(x_{\mathrm{t}})$ for the visual and textual inputs, respectively. These embeddings capture the individual semantic representations of each modality through dedicated encoders. The joint embedding of a vision-language pair $(x_{\mathrm{v}}, x_{\mathrm{t}})$ is defined as $f_\theta(x_{\mathrm{v}}, x_{\mathrm{t}})$, which integrates information from both modalities to support downstream tasks such as classification or retrieval. For fusion-based models (e.g., ALBEF, BLIP), the joint embedding $f_\theta(x_{\mathrm{v}}, x_{\mathrm{t}})$ is obtained from a multimodal fusion encoder, often using the representation of the `[CLS]` token to capture cross-modal interactions. In contrast, dual-encoder models (e.g., CLIP) construct the joint embedding by concatenating the modality-specific features: $f_{\theta_k}(x_{\mathrm{v}}, x_{\mathrm{t}}) = [f_\theta^{\mathrm{v}}(x_{\mathrm{v}}); f_\theta^{\mathrm{t}}(x_{\mathrm{t}})]$.

**Adversarial Example Construction.** Adversarial inputs are generated by adding small perturbations: $\bar{x}_{\mathrm{v}} = x_{\mathrm{v}} + \delta_{\mathrm{v}}, \quad \bar{x}_{\mathrm{t}} = x_{\mathrm{t}} + \delta_{\mathrm{t}}$, where $\delta_{\mathrm{v}}$ denotes a pixel-level modification and $\delta_{\mathrm{t}}$ denotes discrete textual edits. These perturbations are optimized to maximize a task-specific adversarial loss under norm and semantic constraints:

$$(\bar{x}_{\mathrm{v}}, \bar{x}_{\mathrm{t}}) = \arg \max_{(\bar{x}_{\mathrm{v}}, \bar{x}_{\mathrm{t}}) \in \mathcal{S}(x_{\mathrm{v}}, x_{\mathrm{t}})} \mathcal{L}_{\mathrm{adv}}(\mathcal{M}_\theta(\bar{x}_{\mathrm{v}}, \bar{x}_{\mathrm{t}}), y),$$
$$\text{where} \quad \mathcal{S}(x_{\mathrm{v}}, x_{\mathrm{t}}) = \left\{ (\bar{x}_{\mathrm{v}}, \bar{x}_{\mathrm{t}}) \,\middle|\, \|\bar{x}_{\mathrm{v}} - x_{\mathrm{v}}\|_\infty \leq \epsilon_{\mathrm{v}}, \ \bar{x}_{\mathrm{t}} = R(x_{\mathrm{t}}, \epsilon_t) \right\} \tag{1}$$

where $\mathcal{S}(x_{\mathrm{v}}, x_{\mathrm{t}})$ denotes the feasible perturbation space, with visual perturbations constrained by an $\ell_\infty$-norm bound $\epsilon_{\mathrm{v}}$. $R(t)$ denotes the operation of replacing or modifying tokens in the input text and the maximum perturbation $\epsilon_t$ is constrained to the token level. Two representative methods that instantiate Eq. equation 1 are **Co-Attack** (Zhang et al., 2022) and **SGA-Attack** (Lu et al., 2023a). Co-Attack follows a sequential strategy: it first perturbs the text input, then adapts the visual input to exacerbate cross-modal inconsistency. In contrast, SGA-Attack (Set-level Gradient-Aligned Attack) generalizes the adversarial formulation to sets of image-text pairs, aligning gradients at the set level to enhance both transferability and adversarial effectiveness across samples.

**Supervised Adversarial Fine-Tuning.** To improve robustness, adversarial fine-tuning minimizes the loss over adversarial examples. This forms a bi-level optimization:

$$\min_{\theta \in \Theta} \sum_{(x_{\mathrm{v}}, x_{\mathrm{t}}, y) \in \mathcal{D}} \max_{(\bar{x}_{\mathrm{v}}, \bar{x}_{\mathrm{t}}) \in \mathcal{S}(x_{\mathrm{v}}, x_{\mathrm{t}})} \mathcal{L}_{\mathrm{sup}}(\mathcal{M}_\theta(\bar{x}_{\mathrm{v}}, \bar{x}_{\mathrm{t}}), y), \tag{2}$$

where $\mathcal{S}(x_{\mathrm{v}}, x_{\mathrm{t}})$ is the constraints defined in Eq. equation 1. **TeCoA** (Mao et al., 2022) is a special case of this framework where only visual perturbations are allowed ($\epsilon_{\mathrm{t}} = 0$).

**Unsupervised Adversarial Fine-Tuning.** In the absence of ground-truth labels, robustness is promoted by encouraging invariance in representations under adversarial perturbations:

$$\min_{\theta \in \Theta} \sum_{(x_{\mathrm{v}}, x_{\mathrm{t}}) \in \mathcal{D}} \max_{(\bar{x}_{\mathrm{v}}, \bar{x}_{\mathrm{t}}) \in \mathcal{S}(x_{\mathrm{v}}, x_{\mathrm{t}})} \mathcal{L}_{\mathrm{unsup}}(f_{\theta}^{\mathrm{v}}(\bar{x}_{\mathrm{v}}), f_{\theta}^{\mathrm{v}}(x_{\mathrm{v}})), \tag{3}$$

where $\mathcal{L}_{\mathrm{unsup}}$ is typically an $\ell_2$-distance or contrastive loss. **FARE** (Schlarmann et al., 2024) fits into this framework, again with $\epsilon_{\mathrm{t}} = 0$ (visual-only perturbations).

## 4  ADAPTIVE ADVERSARIAL FINE-TUNING

### 4.1  THE MODEL

We propose **Multimodal Adaptive Adversarial Fine-tuning (MAAF)**, a flexible, input-aware defense framework (see Figure 1). Starting from a pre-trained base model $\theta_0$, MAAF learns a set of *defense vectors* $\{\Delta\theta_1, \dots, \Delta\theta_K\}$, each encoding robustness to a specific attack:

$$\Delta\theta_k \triangleq \theta_k - \theta_0, \quad k = 1, \dots, K. \tag{4}$$

A fusion network $\mathcal{F}_\phi$ is learned to predict input-dependent interpolation coefficients:

$$\mathcal{F}_\phi(x_v, x_t) = [\lambda_1, \dots, \lambda_K], \quad \lambda = \mathrm{softmax}(g_\phi(f_{\theta_0}(x_v, x_t))), \tag{5}$$

where $f_{\theta_0}(\cdot)$ is a joint embedding from the base model, and $g_\phi$ is an MLP or attention module. The final input-conditioned model is obtained by linearly combining the defense vectors:

$$\theta(x_v, x_t; \phi, \{\Delta\theta_k\}) = \theta_0 + \sum_{k=1}^{K} \lambda_k(x_v, x_t; \phi) \, \Delta\theta_k. \tag{6}$$

**Task Loss.** For diverse multimodal tasks, we define a unified task loss:

$$\mathcal{L}_{\mathrm{task}}^{\mathcal{T}}(x_v, x_t, y; \theta) = \mathrm{CE}\big(\mathcal{M}_\theta^{(\mathcal{T})}(x_v, x_t), \, y^{(\mathcal{T})}\big), \tag{7}$$

where $\mathcal{M}_\theta^{(\mathcal{T})}$ is the model for the specific task and $y^{(\mathcal{T})}$ is the corresponding supervision signal. For example, in cross-modal retrieval (CR) task, $y^{(\mathrm{CR})} \in \{0, 1\}$ indicates semantic alignment; in visual entailment (VE) task, the output is a three-way softmax over $\{$*entailment*, *neutral*, *contradiction*$\}$; and in visual grounding (VG) task, $y^{(\mathrm{VG})} = (y^{\mathrm{cls}}, y^{\mathrm{box}})$ combines classification and localization targets.

**Target Loss.** The overall loss in MAAF is designed to jointly enforce multiple objectives:

$$\mathcal{L}(x_v, x_t, \bar{x}_v, \bar{x}_t, y; \theta) = \underbrace{\mathcal{L}_{\mathrm{task}}^{\mathcal{T}}(x_v, x_t, y; \theta)}_{\text{task loss on clean data}} + \underbrace{\lambda_{\mathrm{task}} \mathcal{L}_{\mathrm{task}}^{\mathcal{T}}(\bar{x}_v, \bar{x}_t, y; \theta)}_{\text{task loss on adversarial data}}$$
$$+ \underbrace{\lambda_{\mathrm{vl}} \|f_\theta(x_v, x_t) - f_\theta(\bar{x}_v, \bar{x}_t)\|_2^2}_{\text{embedding alignment}} + \lambda_{\mathrm{cos}} \underbrace{\sum_{i,j=1, i \neq j}^{K} \left( \frac{\Delta\theta_i^\top \Delta\theta_j}{\|\Delta\theta_i\| \|\Delta\theta_j\|} \right)^2}_{\text{cosine regularization}}. \tag{8}$$

Here, the first term ensures the model preserves performance on clean inputs, while the second term improves robustness by training the model to make correct predictions under adversarial perturbations. The third term enforces alignment between the embeddings of clean and perturbed inputs, so that small adversarial changes do not dramatically alter the multimodal representations. Finally, the fourth term encourages diversity among the defense vectors, preventing redundancy and allowing each vector to specialize in defending against different types of attacks.

**Bi-level Optimization.** To achieve adaptive robustness against diverse multimodal attacks, MAAF employs a bi-level optimization framework. The bi-level structure explicitly accounts for the worst-case adversarial perturbations while simultaneously learning the defense vectors and the fusion network. Formally, it is defined as

$$
\min_{\phi, \{\Delta\theta_k\}} \mathbb{E}_{(x_v, x_t, y) \sim \mathcal{D}} \Big[ \mathcal{L}\big(x_v, x_t, \bar{x}_v^*, \bar{x}_t^*, y; \theta(\bar{x}_v^*, \bar{x}_t^*; \phi, \{\Delta\theta_k\}))\big) \Big],
$$
$$
\text{s.t. } (\bar{x}_v^*, \bar{x}_t^*) = \arg \max_{(\bar{x}_v, \bar{x}_t) \in \mathcal{S}_r(x_v, x_t)} \mathcal{L}(x_v, x_t, \bar{x}_v, \bar{x}_t, y; \theta(\bar{x}_v, \bar{x}_t; \phi, \{\Delta\theta_k\})).
\tag{9}
$$

In this formulation, the inner maximization finds the strongest perturbation within the threat set $\mathcal{S}_r$, simulating worst-case attacks. The outer minimization then updates the fusion network and defense vectors to minimize the loss under these perturbations. This two-level optimization enables the model to adaptively combine specialized defenses based on input characteristics, achieving robust protection against diverse multimodal attacks.

**Randomized Multimodal Attacks.** Different from existing cross-attack methods that use fixed perturbation budgets $(\epsilon_v, \epsilon_t)$, we propose to randomize these budgets when defining the adversarial search space $\mathcal{S}_r(x_v, x_t)$. This stochastic formulation samples diverse attacks by varying the visual and textual budgets, naturally covering *unimodal visual attack* ($\epsilon_v > 0, \epsilon_t = 0$), *unimodal textual attack* ($\epsilon_v = 0, \epsilon_t > 0$), and *multimodal co-attack* ($\epsilon_v > 0, \epsilon_t > 0$) settings.

### 4.2 Optimization

The bi-level optimization in Equation (9) can be solved via an alternating procedure that combines adversarial input generation with parameter updates. Specifically:

1. **Adversarial Example Generation (Inner Maximization):** We propose an iterative co-attacks strategy to generate multimodal adversarial examples $(\bar{x}_v^*, \bar{x}_t^*)$. Instead of applying visual and textual perturbations independently, we alternate between them for a fixed number of iterations $T$, allowing cross-modal influence: $(\bar{x}_v^*, \bar{x}_t^*) \approx \arg \max_{(\bar{x}_v, \bar{x}_t) \in \mathcal{S}_r(x_v, x_t)} \mathcal{L}(x_v, x_t, \bar{x}_v, \bar{x}_t, y; \theta)$, where $\theta$ is computed according to Equation 6. The iterative procedure proceeds as follows:

   - **Initialization:** Set $(\bar{x}_v^{(0)}, \bar{x}_t^{(0)}) = (x_v, x_t)$.
   - **Repeat for** $t = 0, \ldots, T-1$:
     - Sampling $(\epsilon_v, \epsilon_t)$ from predefined configuration.
     - *Visual Perturbation:* Update $\bar{x}_v^{(t+1)}$ via PGD: $\bar{x}_v^{(t+1)} = \Pi_{\mathcal{S}_v}(\bar{x}_v^{(t)} + \alpha \cdot \nabla_{\bar{x}_v} \mathcal{L}(x_v, x_t, \bar{x}_v^{(t)}, \bar{x}_t^{(t)}, y; \theta))$, where $\alpha > 0$ is the step size, and $\Pi_{\mathcal{S}_v}(\cdot)$ denotes the projection operator that enforces the perturbation constraint $\bar{x}_v^{(t+1)} \in \mathcal{S}_v$. The set $\mathcal{S}_v$ is projection of $\mathcal{S}_r$, which is an $\ell_p$-norm ball with radius $\epsilon_v$ centered at $\bar{x}_v^{(t)}$, ensuring the updated perturbation stays within a bounded neighborhood of the previous iterate.
     - *Textual Perturbation:* At each step $t+1$, we generate a new adversarial text within budget $\epsilon_t$ using BERT-Attack (Li et al., 2020): $\bar{x}_t^{(t+1)} = \text{BERT-Attack}(\bar{x}_t^{(t)}, \epsilon_t \mid \bar{x}_v^{(t+1)}, y; \theta)$, which replaces tokens that maximize the loss while being conditioned on the updated visual perturbation $\bar{x}_v^{(t+1)}$.
   - **Final Output:** Return $(\bar{x}_v^*, \bar{x}_t^*) = (\bar{x}_v^{(T)}, \bar{x}_t^{(T)})$ after $T$ steps.

2. **Parameter Update (Outer Minimization):** Given the adversarial example $(\bar{x}_v^*, \bar{x}_t^*)$, update parameters by minimizing $\min_{\phi, \{\Delta\theta_k\}} \mathcal{L}(x_v, x_t, \bar{x}_v^*, \bar{x}_t^*, y; \theta(\bar{x}_v^*, \bar{x}_t^*; \phi, \{\Delta\theta_k\}))$,. Gradients are computed via backpropagation, and the optimization proceeds using stochastic gradient descent or Adam. As is standard, we do not backpropagate through the adversarial generation steps (Madry et al., 2018b).

## 5 Experiments

### 5.1 Experimental Setup

**Models.** We evaluate two representative categories of vision-language pre-trained models (VLPs): aligned and fused architectures. Aligned VLPs such as CLIP (ViT-B/16)Radford et al. (2021) adopt

a dual-encoder design that encodes images and text separately into a shared embedding space using a contrastive learning objective. In contrast, fused VLPs like ALBEFLi et al. (2021) and TCL (Yang et al., 2022) employ a unified architecture that integrates a Vision Transformer (ViT-B/16) (Dosovitskiy et al., 2021) as the visual encoder with a 6-layer text encoder and a 6-layer multimodal fusion encoder to jointly learn cross-modal representations.

**Benchmarks.** We evaluate on three widely used vision–language benchmarks covering two tasks. For *Image–Text Retrieval*, we use Flickr30K (Plummer et al., 2015) and MSCOCO (Lin et al., 2014). Following common practice, models are fine-tuned on the MSCOCO training set and tested on the Flickr30K test set, and we report Top-1 Text Retrieval (TR@1) and Image Retrieval (IR@1) accuracy. For *Visual Entailment*, we use SNLI-VE (Xie et al., 2019), reporting standard classification accuracy (ACC).

**Baselines.** We compare our proposed method with two existing adversarial fine-tuning approaches. **TeCoA** (Mao et al., 2022) performs text-guided contrastive adversarial training to align robust visual representations. **FARE** (Schlarmann et al., 2024) is an unsupervised approach that encourages adversarial features to stay close to the clean features of the original model. For evaluation, we consider visual perturbations using PGD (Madry et al., 2018a), textual perturbations using BERT-Attack (Li et al., 2020), and multimodal attacks using Co-Attack (Zhang et al., 2022) and SGA (Lu et al., 2023a). Additional implementation details are provided in Appendix .1.

## 5.2 QUANTITATIVE RESULTS

Table 1: Clean and adversarial accuracy on Flickr30k dataset for image-text retrieval task.

| VLPs | Methods | Clean | | BERT-Attack | | PGD | | | | Co-Attack | | | |
|---|---|---|---|---|---|---|---|---|---|---|---|---|---|
| | | | | | | 2/255 | | 4/255 | | 2/255 | | 4/255 | |
| | | TR↑ | IR↑ | TR↑ | IR↑ | TR↑ | IR↑ | TR↑ | IR↑ | TR↑ | IR↑ | TR↑ | IR↑ |
| CLIP_ViT | Origin | **81.5** | 62.1 | 61.7 | 41.1 | 27.5 | 16.4 | 9.4 | 7.9 | 8.3 | 4.2 | 1.5 | 0.5 |
| | TeCoA[2] | 81.0 | 60.0 | 53.2 | 32.8 | 53.7 | 32.2 | 29.5 | 20.2 | 22.3 | 11.6 | 18.3 | 9.7 |
| | FARE[2] | 81.0 | 62.6 | 54.6 | 34.3 | 52.9 | 31.3 | 24.6 | 15.1 | 30.0 | 16.9 | 12.8 | 7.7 |
| | MAAF[2] | 81.0 | **66.1** | **71.7** | **52.2** | **75.4** | **59.7** | **65.8** | **52.0** | 46.8 | 32.5 | 57.8 | 41.2 |
| | TeCoA[4] | 76.7 | 59.3 | 49.7 | 30.6 | 60.2 | 41.2 | 42.9 | 28.6 | 29.8 | 17.4 | 16.6 | 10.2 |
| | FARE[4] | **81.0** | 64.7 | 53.2 | 32.0 | 61.7 | 40.2 | 33.6 | 21.8 | 26.0 | 13.4 | 10.4 | 4.5 |
| | MAAF[4] | 77.0 | **65.4** | **68.8** | **52.3** | **73.1** | **60.5** | **68.8** | **53.3** | 57.2 | 42.8 | 47.1 | 34.1 |
| ALBEF | Origin | **94.9** | 84.5 | 81.4 | 69.1 | 31.9 | 23.6 | 12.0 | 9.3 | 27.3 | 23.5 | 13.0 | 11.9 |
| | TeCoA[2] | 93.0 | **85.7** | 64.5 | 45.3 | 50.1 | 39.8 | 23.7 | 19.9 | 34.8 | 27.9 | 16.1 | 14.7 |
| | FARE[2] | 94.6 | 84.5 | 70.1 | 49.8 | 80.0 | **66.5** | **64.4** | 50.8 | 74.3 | 53.7 | 62.8 | 46.2 |
| | MAAF[2] | 94.1 | 84.7 | **82.7** | **70.4** | **86.6** | 61.3 | 63.9 | **51.2** | 79.2 | 54.6 | 68.4 | 47.0 |
| | TeCoA[4] | 94.0 | 84.5 | 68.2 | 48.5 | 47.7 | 39.0 | 23.2 | 19.3 | 32.9 | 23.4 | 16.9 | 12.6 |
| | FARE[4] | 90.7 | 80.2 | 73.0 | 51.1 | 82.7 | 71.9 | **81.4** | 70.6 | 75.7 | 53.9 | 70.8 | **50.3** |
| | MAAF[4] | 93.6 | 81.9 | **80.2** | **69.0** | **85.5** | **75.4** | **84.7** | **72.5** | 82.1 | 65.9 | 75.6 | 51.8 |
| TCL | Origin | **94.9** | 80.4 | 75.7 | 64.5 | 42.2 | 30.2 | 18.8 | 11.8 | 13.5 | 13.0 | 2.8 | 3.1 |
| | TeCoA[2] | 90.9 | **84.7** | 62.8 | 50.4 | 56.7 | 43.1 | 40.1 | 29.3 | 28.1 | 26.4 | 8.2 | 9.3 |
| | FARE[2] | 93.1 | 82.5 | 71.5 | 59.6 | 78.4 | **65.8** | 64.0 | 51.3 | 57.4 | 25.2 | 32.9 | **33.5** |
| | MAAF[2] | 94.1 | 84.0 | **79.7** | **71.2** | **80.9** | 65.7 | 64.8 | 51.9 | 60.7 | 47.9 | **42.0** | 29.5 |
| | TeCoA[4] | 94.3 | 83.5 | 60.9 | 49.1 | 57.1 | 44.2 | 38.6 | 30.1 | 17.1 | 14.1 | 5.9 | 7.2 |
| | FARE[4] | 94.0 | 82.9 | 73.6 | 60.4 | 76.5 | 62.4 | 65.0 | 51.6 | 56.6 | 39.7 | 34.6 | 25.9 |
| | MAAF[4] | 94.1 | 80.6 | **79.2** | **66.7** | **83.8** | **74.0** | **80.9** | **68.2** | 60.4 | 46.5 | 46.4 | **30.9** |

**Performance on Image-Text Retrieval.** Table 1 presents the clean and adversarial accuracy of vision-language models on the Flickr30k image-text retrieval benchmark. Across all backbones—CLIP_ViT, ALBEF, and TCL—our method MAAF consistently achieves the best adversarial robustness while maintaining strong clean performance. Under the strongest attack setting, Co-Attack ($\epsilon = 4/255$), MAAF[4] significantly outperforms existing defenses. For CLIP_ViT, MAAF[4] achieves **47.1%** text retrieval (TR) and **34.1%** image retrieval (IR), compared to only 16.6/10.2 from TeCoA[4] and 10.4/4.5 from FARE[4]—representing improvements of 30.5% in TR and 23.9% in IR over the strongest baseline. On ALBEF, MAAF[4] achieves **75.6%** TR and **51.8%** IR under Co-Attack

(4/255), substantially higher than FARE[4] (70.8/50.3) and TeCoA[4] (16.9/12.6). Notably, MAAF preserves clean accuracy well: ALBEF with MAAF[4] retains 93.6% TR and 81.9% IR. Similarly, on the TCL backbone, MAAF[4] achieves **46.4%** TR and **30.9%** IR under Co-Attack (4/255), far surpassing FARE[4] (34.6/25.9) and TeCoA[4] (5.9/7.2), while maintaining high clean performance (94.1% TR, 80.6% IR). MAAF also excels under other attacks: under PGD (4/255), MAAF[4] yields 68.8/53.3 on CLIP$_{ViT}$, 84.7/72.5 on ALBEF, and 80.9/68.2 on TCL—consistently outperforming all baselines.

Table 2: Clean and adversarial accuracy on SNLI-VE dataset for visual entailment.

| VLPs | Adversarial fine-tuning | Clean | BERT-Attack | PGD | | Co-Attack | |
|---|---|---|---|---|---|---|---|
| | | | | 2/255 | 4/255 | 2/255 | 4/255 |
| ALBEF | Origin | 83.3 | 70.4 | 32.8 | 24.5 | 21.9 | 17.5 |
| | TeCoA[2] | **83.6** | 62.0 | 47.4 | 28.9 | 25.2 | 19.1 |
| | FARE[2] | 81.8 | 71.9 | 64.6 | 54.2 | 35.1 | 28.2 |
| | MAAF[2] | 83.1 | **74.5** | **68.3** | **56.4** | **38.2** | **34.5** |
| | TeCoA[4] | **83.5** | 61.3 | 45.6 | 29.0 | 24.1 | 18.6 |
| | FARE[4] | 82.7 | 69.8 | 58.4 | 50.2 | 31.4 | 24.5 |
| | MAAF[4] | 83.2 | **72.6** | **61.6** | **54.2** | **32.8** | **28.0** |
| TCL | Origin | 79.3 | 64.9 | 38.6 | 28.4 | 21.0 | 18.9 |
| | TeCoA[2] | **79.8** | 59.4 | 43.5 | 30.9 | 23.9 | 20.1 |
| | FARE[2] | 77.2 | 63.4 | 50.2 | 36.8 | 30.3 | 27.2 |
| | MAAF[2] | 78.5 | **69.3** | **60.8** | **45.4** | **34.6** | **28.1** |
| | TeCoA[4] | **79.8** | 59.7 | 44.1 | 32.0 | 23.8 | 20.0 |
| | FARE[4] | 79.0 | 61.1 | 48.6 | 32.5 | 26.2 | 21.6 |
| | MAAF[4] | 79.0 | **65.5** | **56.8** | **42.7** | **28.4** | **23.8** |

**Performance on Visual Entailment.** As shown in Table 2, the visual entailment task on SNLI-VE further demonstrates the robustness of MAAF. The original ALBEF model suffers a sharp accuracy drop from **83.3%** to **17.5%** under Co-Attack ($\epsilon = 4/255$), revealing its vulnerability to multimodal adversarial perturbations. In contrast, **MAAF[2] maintains high clean accuracy** (**83.1%**) while significantly improving robustness across all attacks: it achieves **74.5%** under BERT-Attack, **68.3%/56.4%** under PGD (2/255 and 4/255), and **38.2%/34.5%** under Co-Attack—consistently outperforming baselines. On the TCL backbone, MAAF also achieves the best performance in every adversarial setting without sacrificing clean accuracy (78.5% for MAAF[2], 79.0% for MAAF[4]).

Table 3: Transfer-based attack results on Flickr30K with ALBEF as surrogate and TCL as target.

| VLPs | Adversarial fine-tuning | Clean | | SGA | | | | | | | |
|---|---|---|---|---|---|---|---|---|---|---|---|
| | | | | 2/255 | | | | 4/255 | | | |
| | | TR@1 | IR@1 | TR@1 | TR@5 | IR@1 | IR@5 | TR@1 | TR@5 | IR@1 | IR@5 |
| TCL | Origin | **94.9** | 84.0 | 51.1 | 75.0 | 38.2 | 62.0 | 33.7 | 56.8 | 26.5 | 47.3 |
| | TeCoA[2] | 90.9 | **84.7** | 48.4 | 67.3 | 42.7 | 66.2 | 35.0 | 52.5 | 31.5 | 54.3 |
| | FARE[2] | 93.1 | 82.5 | 61.4 | 82.9 | 45.2 | **77.8** | 46.8 | 70.3 | 35.2 | 58.9 |
| | MAAF[2] | 94.1 | 84.0 | **63.7** | **83.0** | **46.9** | 71.2 | **49.3** | **75.5** | **37.6** | **59.8** |
| | TeCoA[4] | 88.8 | **83.5** | 44.9 | 62.9 | 41.9 | 65.9 | 31.5 | 48.6 | 30.9 | 53.2 |
| | FARE[4] | 94.0 | 82.9 | 59.9 | 82.1 | 45.0 | 69.0 | 46.4 | 71.0 | 35.4 | 58.9 |
| | MAAF[4] | **94.1** | 80.6 | **62.4** | **83.6** | 45.2 | 70.4 | **48.9** | **74.5** | **38.0** | **60.8** |

**Transfer-based Attack Evaluation.** We evaluate robustness using transfer-based attacks, where adversarial examples generated from one model (the original ALBEF) are applied to a different target model (TCL) on the Flickr30K test set. This simulates a realistic scenario in which attackers can access only a surrogate model, not the target model's internal parameters. We also include the Set-level Guidance Attack (SGA) (Lu et al., 2023b), which exploits set-level cross-modal interactions and substantially improves attack transferability. As shown in Table 3, the original TCL model is highly vulnerable to transferred adversarial examples. Under an SGA attack with $\epsilon = 4/255$, its TR1 drops sharply from 94.9% to 33.7%, and IR1 falls from 84.0% to 26.5%. Fine-tuning with TeCoA improves robustness slightly (TR1: 35.0%, IR1: 31.5%) but sacrifices clean accuracy. FARE

achieves stronger robustness (TR1: 46.8%, IR1: 35.2%) but still leaves significant vulnerability. In contrast, MAAF[2] consistently achieves the best trade-off. It maintains high clean accuracy (94.1%) and significantly improves robustness, achieving TR1 of 49.3% and IR1 of 37.6% under $\epsilon = 4/255$. MAAF outperforms both TeCoA and FARE in terms of clean and robust accuracy, demonstrating its superior generalization under challenging transferred multimodal adversarial attacks.

Table 4: Ablation results of MAAF on Flickr30K for the image–text retrieval task using $\text{CLIP}_{\text{ViT}}$.

| Methods | Clean | | BERT-Attack | | PGD | | Co-Attack | |
|---|---|---|---|---|---|---|---|---|
| | TR | IR | TR | IR | TR | IR | TR | IR |
| $\text{MAAF}^4_{w/o\ \mathcal{L}_{vl}}$ | 68.2 | 49.6 | 44.9 | 25.8 | 48.8 | 27.9 | 25.9 | 18.6 |
| $\text{MAAF}^4_{w/o\ \mathcal{L}_{\cos}}$ | 81.0 | 60.3 | 52.4 | 31.0 | 49.5 | 26.6 | 24.4 | 18.0 |
| $\text{MAAF}^4_{w/o\ \mathcal{L}_{vl}, \mathcal{L}_{\cos}}$ | 61.4 | 45.8 | 44.6 | 23.5 | 46.9 | 24.6 | 20.8 | 16.7 |
| $\text{MAAF}^4$ | **77.0** | **65.4** | **68.8** | **52.3** | **68.8** | **53.3** | **47.1** | **34.1** |

**Ablation Studies.** To assess the contribution of key components in our framework, we conduct ablation experiments on the $\text{CLIP}_{\text{ViT}}$ model for image–text retrieval. As shown in Table 4, each term in our overall loss (Eq. 8) plays a distinct and vital role. Removing the embedding alignment term $\mathcal{L}_{vl} = \|f_\theta(x_v, x_t) - f_\theta(\bar{x}_v, \bar{x}_t)\|_2^2$ (i.e., $\text{MAAF}^4_{w/o\ \mathcal{L}_{vl}}$) significantly degrades both clean and robust performance—clean accuracy drops to 68.2/49.6 (TR/IR), and under Co-Attack ($\epsilon = 4/255$), it falls to 25.9/18.6. This confirms that aligning clean and adversarial multimodal embeddings is crucial for preserving semantic consistency under perturbations. Ablating the cosine regularization term $\mathcal{L}_{\cos} = \sum_{i \neq j} \left(\frac{\Delta\theta_i^\top \Delta\theta_j}{\|\Delta\theta_i\|\|\Delta\theta_j\|}\right)^2$ ($\text{MAAF}^4_{w/o\ \mathcal{L}_{\cos}}$) also weakens robustness, particularly under BERT-Attack (52.4/31.0) and Co-Attack (24.4/18.0), despite maintaining relatively high clean accuracy (81.0/60.3). This indicates that encouraging diversity among defense directions is essential for handling heterogeneous attack types. When both terms are removed, performance deteriorates further across all settings (e.g., 20.8/16.7 under Co-Attack), highlighting their complementary nature. In contrast, the full $\text{MAAF}^4$ model achieves the best results in all scenarios, with 77.0/65.4 clean accuracy and 47.1/34.1 under Co-Attack (4/255). These results validate that both the embedding alignment and cosine regularization terms are indispensable for achieving strong and versatile multimodal adversarial robustness.

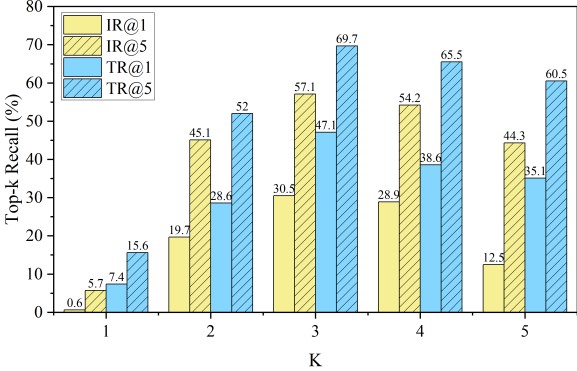

Figure 2: Comparison of $\text{CLIP}_{\text{ViT}}$ on Flickr30K fine-tuned with varying numbers of defense vectors $K$ for the image-text retrieval task.

**Sensitivity to the Number of Defense Vectors $K$.** Figure 2 shows the Top-$k$ recall (%) of CLIP on the image-text retrieval task under the attack configuration space $\mathcal{E} = \left\{0, \frac{2}{255}, \frac{4}{255}\right\} \times \{0, 1\} \setminus \{(0, 0)\}$, with the number of defense vectors $K$ ranging from 1 to 6. We find that increasing $K$ generally improves robustness, with performance peaking at $K = 3$. At this setting, IR@5 and TR@5 reach 41.6% and 28.5%, respectively, suggesting that moderate adversarial diversity during fine-tuning enhances generalization against multimodal attacks. Beyond $K = 3$, improvements plateau or slightly decrease, likely due to gradient conflicts among overly heterogeneous vectors. This pat-

tern is consistent across both retrieval directions and recall levels, indicating that a controlled number of defense vectors balances specialization and generalization effectively.

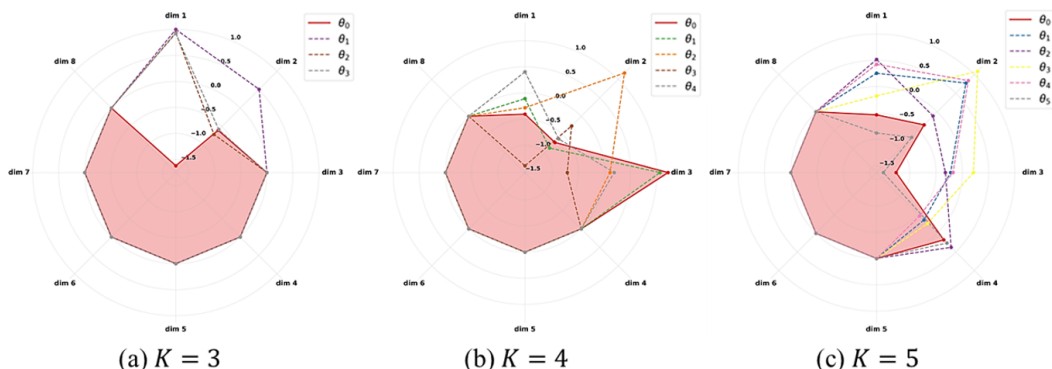

(a) $K = 3$  (b) $K = 4$  (c) $K = 5$

Figure 3: PCA visualization of defense vectors for $CLIP_{ViT}$ on Flickr30K across different $K$.

**Visualization of Defense Vectors.** Figure 3 shows a PCA projection of the defense vector embeddings into 8 dimensions, illustrating how their distribution varies with the number of adversarial sources $K$. For $K = 4$, the vectors appear more widely dispersed in the embedding space compared to $K = 3$, potentially offering broader coverage of adversarial directions. However, as shown in Figure 2, $K = 3$ achieves better performance, indicating that while a more spread-out vector set may seem desirable, excessive dispersion can introduce conflicts or instability, ultimately reducing adaptive effectiveness.

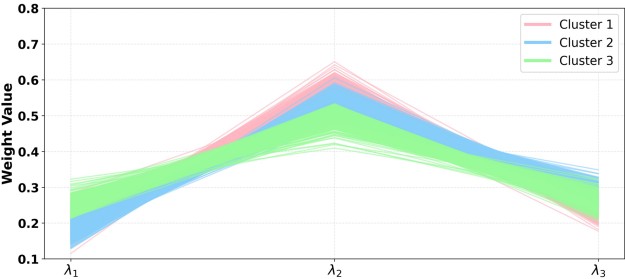

Figure 4: Fusion weights distribution of $CLIP_{ViT}$ for samples on Flickr30K with $K = 3$.

**Distribution of Input-Dependent Fusion Weights.** Figure 4 visualizes the input-dependent fusion weights predicted by $\mathcal{F}_\phi$ on Flickr30K samples with $K = 3$. The weights vary across samples, highlighting the need to adaptively assemble a model for each input. They form three loosely defined clusters, reflecting subtle differences between input groups. Importantly, all samples assign the largest weight to the second defense vector, emphasizing its key role in robust performance.

## 6 CONCLUSION

In this paper, we propose MAAF, a novel defense framework that generates input-conditioned model parameters to protect against diverse and unseen adversarial attacks. Extensive experiments show that MAAF significantly enhances adversarial robustness across multiple attack types while maintaining clean accuracy, and provide several insights into adaptive model synthesis. Despite these advantages, MAAF introduces additional inference overhead due to dynamic model synthesis and requires careful tuning of fusion and optimization components. Future work will focus on developing more efficient architectures for input-conditioned adaptation, extending the framework to real-world multimodal corruptions, and applying MAAF to a broader set of multimodal tasks and architectures.

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

## .1 IMPLEMENTATION DETAILS

All experiments are conducted in PyTorch. Models are adversarially fine-tuned for 10 epochs with our MAAF algorithm using a batch size of $B = 64$ and $K = 3$ defense vectors. The regularization weights are set to $\lambda_{\text{task}} = \lambda_{\text{vl}} = \lambda_{\text{cos}} = 1$, and we run $T = 10$ attack iterations during training. Adam is employed with learning rates $\eta_\theta = 2\times10^{-5}$ for the base model parameters and $\eta_\phi = 1\times10^{-4}$ for the fusion network $\mathcal{F}_\phi$; the pre-trained base parameters $\theta_0$ are kept fixed.

For adversarial training and evaluation, we define the attack space $\mathcal{E} = (\epsilon_v, \epsilon_t) \in \{0, \frac{2}{255}, \frac{4}{255}\} \times \{0, 1\} \setminus \{(0,0)\}$, where $\epsilon_v$ is the visual perturbation budget and $\epsilon_t$ indicates whether a textual attack is applied via single-word substitution. Visual perturbations are generated by PGD (Madry et al., 2018a) with $\epsilon_v \in \{2/255, 4/255\}$, step size $\alpha = 0.5/255$, and $T = 10$ iterations. Textual perturbations are produced by BERT-Attack (Li et al., 2020) with $\epsilon_t = 1$. Both Co-attack (Zhang et al., 2022) and SGA (Lu et al., 2023a) are with $\epsilon_v \in \{2/255, 4/255\}$ and $\epsilon_t = 1$. In all reported results, superscripts denote the visual perturbation budget: for example, MAAF[2], TeCoA[2], and FARE[2] correspond to $\epsilon_v = 2/255$, while the superscript 4 refers to $\epsilon_v = 4/255$; in each case, $\epsilon_t \in \{0, 1\}$ is determined by the attack type.

## .2 SENSITIVITY TO $\lambda$

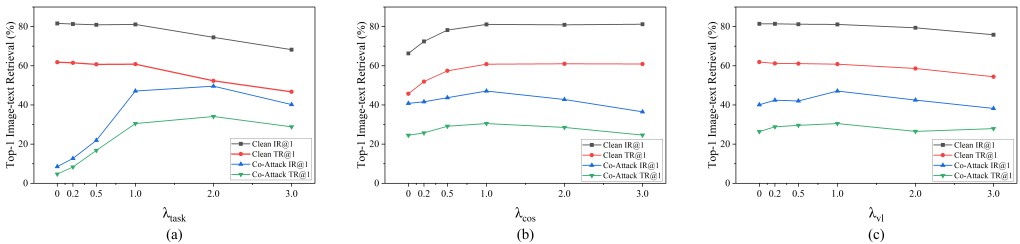

(a)  (b)  (c)

Figure 5: Sensitivity of CLIPViT to $\lambda_{\mathrm{task}}$, $\lambda_{\mathrm{cos}}$, and $\lambda_{\mathrm{vl}}$ on the Flickr30K image–text retrieval task, with all other hyperparameters fixed at 1.0.

Figure 5 shows the Top-1 recall (%) of CLIP on image–text retrieval under the co-attack space $\mathcal{E} = \left\{0, \frac{2}{255}, \frac{4}{255}\right\} \times \{0, 1\} \setminus \{(0, 0)\}$, while varying one hyperparameter at a time and fixing the others to 1.0. Increasing $\lambda_{\mathrm{task}}$ keeps clean retrieval performance (IR@1 and TR@1) nearly unchanged but clearly lowers co-attack robustness, indicating that over-emphasizing the task loss causes adversarial overfitting. Raising $\lambda_{\mathrm{cos}}$ improves both clean and robust performance up to about 1.0, after which the gain plateaus, showing that promoting diversity in parameter updates strengthens robustness but yields diminishing returns when too large. Finally, larger $\lambda_{\mathrm{vl}}$ consistently enhances co-attack robustness with only minor effects on clean accuracy, confirming that enforcing alignment consistency across modalities stabilizes the model. Overall, moderate settings (around 1.0) for all three hyperparameters achieve the best trade-off between clean performance and robustness, balancing task fidelity, update diversity, and multimodal alignment.

