# OpenReview forum: "Towards Robust Multimodal Learning via Adaptive Model Assembly"
_ICLR.cc/2026/Conference — Submitted to ICLR 2026_

### Official Review · Reviewer_MzE1 · 2025-10-27

**Soundness:** 2
**Presentation:** 2
**Contribution:** 2
**Rating:** 2
**Confidence:** 4

**Summary:**

This paper proposed multimodal adaptive adversarial fine-tuning to enhance the adversarial robustness of vision-language pretraining models against unimodal and multimodal attack strategies.
Traditional adversaial fine-tuning methods is designed for unimodal adversarial attacks and cannot be generalized to unseen attacks.

**Strengths:**

Compared with previous adversarial training methods for vision-language models that primarily focus on visual attacks, MAAF explores adversarial robustness by learning a set of defense vectors and dynamically fusing them to construct input-specific defense models, thereby achieving strong robustness against diverse adversarial perturbations.

**Weaknesses:**

1. Insufficient experiments. Adversarial fine-tuning for vision-language pretrained models should consider not only unimodal but also multimodal attack scenarios. The authors claim that the proposed MAAF enhances robustness against both types of attacks. However, the experimental evaluation is incomplete — it omits key benchmarks such as AutoAttack (including APGD with cross-entropy loss and APGD with DLR loss, each with 100 iterations). In addition, the perturbation budget for BERT-Attack is fixed at 1, with no further investigation of different perturbation magnitudes.

2. To ensure the integrity of the experiments, the authors should evaluate MAAF using different vision encoder variants, such as ViT-L/14 and ViT-B/32. In particular, for the transfer-based attack evaluation, it would be important to examine whether MAAF can provide stronger adversarial defenses when applied to multimodal large language models (e.g., LLaVA).


3. The robustness improvement of MAAF may partly result from its use of randomized multimodal attacks during fine-tuning, giving it a stronger and more diverse training setup than the fixed-attack baselines, thus raising fairness concerns in the comparison.

4. The ablation studies are incomplete and do not isolate the key sources of improvement. In particular, the paper lacks experiments that remove the adaptive fusion network, fix the attack distribution, or compare multiple defense vectors without adaptive weighting. As a result, it remains unclear whether the performance gains stem from the proposed adaptive fusion mechanism, the diversity of defense vectors, or the randomized attack training.

5. The paper admit that "MAAF introduces additional inference overhead due to dynamic model synthesis” but lacks quantitative analysis of its computational or inference cost, making it unclear how efficient or scalable MAAF is in practice.

**Questions:**

The questions are detailed in the Weaknesses section above.

---

### Official Review · Reviewer_B4hu · 2025-10-27

**Soundness:** 2
**Presentation:** 2
**Contribution:** 2
**Rating:** 2
**Confidence:** 4

**Summary:**

The paper proposes MAAF, a new framework to defend VLPs against adversarial attacks. Instead of training one static model, MAAF learns a base model and a set of "defense vectors". At inference time, a small fusion network predicts how to best combine these vectors based on the specific input sample. This creates a custom, input-specific model to defend against attacks

**Strengths:**

The core idea of creating an input-specific model assembly is interesting. The loss function (Eq. 8) is well-motivated, especially the inclusion of the embedding alignment term ($\mathcal{L}_{vl}$) and the cosine regularization term ($\mathcal{L}_{cos}$) to ensure semantic consistency and defense diversity.

The authors validate their method across multiple VLP architectures (dual-encoder CLIP and fusion-based ALBEF/TCL) and different tasks (Image-Text Retrieval and Visual Entailment)

**Weaknesses:**

The most importantly, the paper completely fails to cite or discuss a highly relevant and recent category of methods: test-time adversarial prompt tuning [1,2]. It seems that the authors overlook the set of works for adversarial prompt tuning on VLMs.

[1] TAPT: Test-Time Adversarial Prompt Tuning for Robust Inference in Vision-Language Models, CVPR 2025
[2] Clip is strong enough to fight back: Test-time counterattacks towards zero-shot adversarial robustness of clip. CVPR 2025

This omission is the most severe problem. MAAF's core mechanism, adapting parameters at inference time to fight an attack, is conceptually very similar to test-time prompt tuning. In MAAF, a fusion network $\mathcal{F}_{\phi}$ generates weights $\lambda_k$ to create new parameters $\theta$. In TAPT, parameters (the prompts) are also optimized or generated at test time for the same goal. By ignoring this entire field, the paper's claim to novelty is significantly undermined. The method must be compared against these TAPT baselines.

The method introduces significant overhead at inference time. For every sample, the model must first run the fusion network $\mathcal{F}_{\phi}$ and then perform parameter assembly ($\theta_0 + \sum \lambda_k \Delta\theta_k$) before it can even begin the standard forward pass. The authors admit this overhead in the conclusion 16 but provide zero quantitative analysis (e.g., latency in milliseconds, or extra FLOPS). Without this data, the practical utility of the method is unknown.


Minors:

- Image Quality: All figures are low-resolution bitmaps and are blurry. These should be high-quality vector graphics (e.g., PDF/SVG).

- Appendix: The appendix sectioning appears to be incorrect (e.g., starting with .1, .2).

- Missing LLM Declaration: The paper is missing the mandatory ICLR policy statement on the use of Large Language Models (LLMs) for writing or polishing the text, if the authors use it.

**Questions:**

- How does the method fundamentally differ from test-time adversarial prompt tuning (TAPT) methods [1, 2]?Please provide a detailed comparison.
- What is the exact inference overhead of MAAF? Please provide the wall-clock latency (in ms) and/or FLOPS required for the fusion network and parameter assembly, and compare it to the baseline model.
- Why does performance degrade when $K > 3$? Does this mean MAAF can only effectively handle 2-3 "types" of attacks?
- Please add the mandatory ICLR LLM usage declaration if the authors use it.

---

### Official Review · Reviewer_qfa1 · 2025-10-31

**Soundness:** 3
**Presentation:** 3
**Contribution:** 3
**Rating:** 4
**Confidence:** 4

**Summary:**

In this paper, the authors proposed MAAF (Multimodal Adaptive Adversarial Fine-tuning) which is an adversarially trained defense mechanism that is meant to adapt to various attacks during inference. Specifically, during the inner loop of adversarial training, MAAF allows attacks to be sampled from a rather large search space (including ) and it learns a collection of defense vectors (the same size of the entire model), whose linear combination is believed to generate defenses against different types of attacks. MAAF was evaluated on 3 VLP models on image-text retrieval and visual entailment against TeCoA and FARE under attacks of various types. The results show that MAAF consistently achieves better robustness while maintaining clean  accuracy.

**Strengths:**

+ The choice of VLM architectures and tasks for evaluation is pretty comprehensive.
+ MAAF shows strong and consistent empirical improvement over the baselines.
+ Input-adaptive defense is an interesting topic.
+ Good insights from PCA visualization of defense vectors and fusion weight distributions.

**Weaknesses:**

+ The size of each defense vector is the same as the model to protect, which requires the resources to train a model that is multiple times larger than that model, which doesn't sound like a scalable approach when the VLPs are no longer as small as less than 100 million parameters. In a sense MAAF sounds like a MoE but uses linear combination instead of routing.
+ The idea to defend against more attacks by using a more diverse adversarial search space is pretty natural and people have started to do similar staff years ago like in [1] and making defense mechanism to adapt to inputs is also not entirely new as we have test-time adaption or meta-learning based adversarial training like [2].
+ The paper does not include an ablation study to see what if we use a larger adversarial search space but keep the vanilla adversarial training what would be the result, or if we have models trained for individual attack types and then ensemble them what would be the result. The current comparisons with baselines don't in fact validate the merit of MAAF because it already uses more data and more parameters.

1. Tramer, Florian, and Dan Boneh. "Adversarial training and robustness for multiple perturbations." Advances in neural information processing systems 32 (2019).
2. Bamdad, Amirmohammad, Ali Owfi, and Fatemeh Afghah. "Adaptive Meta-learning-based Adversarial Training for Robust Automatic Modulation Classification." arXiv preprint arXiv:2501.01620 (2025).

**Questions:**

Please refer to the weaknesses for my concerns about his paper. The following questions are only listed for easier understanding of what I am concerning.
+ What is the cost of training and using MAAF vs. the unprotected model and the baselines?
+ How would MAAF compare with meta learning based adaptive adversarial training?
+ Can you apply defense vectors only to some select layers of the model?
+ Do attacks of the same type triggers similar defenses?
+ How does MAAF compare to an ensemble of models trained individually on their own search space? Or a model that doesn't uses the defense vectors?

---

### Official Review · Reviewer_3kNM · 2025-11-01

**Soundness:** 2
**Presentation:** 3
**Contribution:** 2
**Rating:** 4
**Confidence:** 5

**Summary:**

This paper proposes MAAF (Multimodal Adaptive Adversarial Fine-tuning), a framework for improving the robustness of vision–language pre-trained models (VLPs) such as CLIP, ALBEF, and TCL. Unlike traditional fine-tuning approaches that produce a single static model, MAAF learns multiple defense vectors and adaptively assembles them at inference time using an input-aware generation network. This design allows for sample-specific parameter adaptation without retraining. Experiments on standard benchmarks show consistent robustness improvements over existing fine-tuning methods, with additional analyses on the behavior and diversity of defense vectors.

**Strengths:**

1. The proposed idea is conceptually interesting. The adaptive assembly of model parameters for each input provides a novel and flexible approach to robustness.

2. The experimental design includes results demonstrating the improvements on multiple VLPs brought by MAAF.

3. The paper includes clear writing and presentation. The paper is well-organized, and the analysis of defense vector distribution adds interpretability and depth.

4. The proposed method shows balanced performance. The work maintains strong clean accuracy while improving adversarial robustness, which is practically valuable.

**Weaknesses:**

1. The attackcoverage is limited. The paper does not evaluate against representative VLM-based attack frameworks such as VLAttack, nor against black-box text adversarial attacks like TextHoaxer (AAAI 2021), LeapAttack (KDD 2022), or PAT (KDD 2023), which limits the generality of the claimed robustness.

2. There is a lack of computational analysis. The cost of adaptive model assembly, including parameter generation and adaptive fusion, is not quantified, making it unclear how scalable the approach is for real-time or large-scale applications.

**Questions:**

1. Could the authors include evaluations under VLAttack and text-based adversarial settings to demonstrate robustness across modalities?

2. How significant is the computational overhead introduced by the adaptive assembly mechanism during inference?

3. Can the proposed method generalize to cross-modal or multimodal black-box attacks?

---

### Meta-Review · Area_Chair_zAAW · 2026-01-07

**Summary:**

Overall, all the reviewers suggest rejecting this paper. The major weaknesses are summarized as follows:
- Insufficient experiments. Reviewer 3kNM, B4hu and MzE1 point out that some important baselines are not compared in this work. Reviewer qfa1 and MzE1 demonstrate that this paper lacks certain ablation studies.
- Lack of overhead analysis. All the reviewers point out that the proposed method might increase the computational overhead while the analysis about overhead is not conducted in the paper.
- Novelty. Reviewer qfa1 and B4hu question the novelty of this paper due to the core method is a general idea with limited innovation.

**Reviewer Concerns:**

The authors provided no response, thus failing to address any of the reviewer concerns.

**Reviewer Scores:**

All the reviewer scores would have remained unchanged as the authors did not participate in the rebuttal.

---

### Decision · Program_Chairs · 2026-01-26

Reject